# Analysis of Deaths among HIV-Infected Patients Hospitalized in 2009–2018 in Main Centre of Infectious Disease in Region of Lower Silesia in Poland, Detailing Lesions in the Central Nervous System

**DOI:** 10.3390/medicina58020270

**Published:** 2022-02-11

**Authors:** Justyna Janocha-Litwin, Aleksander Zińczuk, Sylwia Serafińska, Anna Szymanek-Pasternak, Krzysztof Simon

**Affiliations:** 1Department of Infectious Diseases and Hepatology, Medical University Wroclaw, 50-367 Wroclaw, Poland; alek.zinczuk@gmail.com (A.Z.); sylwia.serafinska@umed.wroc.pl (S.S.); anna.szymanek-pasternak@umed.wroc.pl (A.S.-P.); krzysimon@gmail.com (K.S.); 2Department of Infectious Disease, Provincial Hospital Gromkowskiego, Koszarowa 5, 51-149 Wroclaw, Poland; 3Department of Forensic Medicine, Medical University Wroclaw, 50-367 Wroclaw, Poland

**Keywords:** PLWH, HIV, death, central nervous system, toxoplasmosis

## Abstract

*Background and Objectives*: Patients living with HIV (PLWH), especially those diagnosed too late or not receiving treatment with antiretroviral drugs in the stage of advanced immunodeficiency AIDS for various reasons, develop additional opportunistic infections or AIDS-defining diseases that may contribute directly to the death of these patients. *Material and Methods*: In this work, we focused on disorders of the central nervous system (CNS) by retrospectively analyzing the symptoms, clinical and autopsy diagnoses of patients diagnosed with HIV infection who died in the provincial specialist hospital in the Lower Silesia region in Poland. *Results*: The autopsy was performed in 27.4% cases. The cause of death was determined to be HIV-related/AIDS-associated in 78% patients. The most common AIDS-defining CNS diseases in our cohort were toxoplasmosis and cryptococcosis. *Conslusions*: The presented results of the most common causes of changes in the central nervous system among deceased HIV-infected patients are comparable to the results of studies by other scientists cited in the publication.

## 1. Introduction

The central nervous system (CNS) is the second-most commonly affected organ system (after the respiratory system) by various manifestations secondary to HIV infection and/or AIDS [1]. Therefore, the onset of neurological symptoms, or even a suspected CNS pathology in an HIV-positive patient, should prompt a comprehensive diagnostic evaluation for infectious and other (non-infectious) diseases, including those unrelated to HIV infection (see Table 1) [2,3].

On the other hand, any finding of CNS lesions, in particular in younger patients, should prompt HIV testing, as neurological symptoms can and often are the first manifestation of HIV infection.

The most frequently observed neurological symptoms in HIV patients are: disorientation, conduct disorder, impaired consciousness, seizures, muscle weakness, hemiparesis, headaches concomitant with fever, and positive meningeal signs. All above are the symptoms of CNS pathologies seen mostly in HIV-positive patients and patients with AIDS: toxoplasmosis, cryptococcosis, progressive multifocal leukoencephalopathy (PML), and CNS lymphoma [4].

As in any other case, the diagnostic assessment should begin with a physical examination including a comprehensive neurological assessment. Neuroimaging techniques such as computed tomography and/or magnetic resonance imaging (both with contrast) are crucial for the diagnosis. The literature emphasizes the importance of radiologist experience and expertise in assessing lesions typical of different CNS disorders in HIV-infected patients [5]. The requested laboratory tests should include in each case the CD4 count, HIV RNA viral load (VL), and serology assays for toxoplasmosis, cytomegalovirus, and syphilis. In the absence of generally accepted contraindications, a lumbar puncture should also be performed, followed by cerebrospinal fluid (CSF) analysis and PCR testing to identify possible causal pathogens (e.g., Toxoplasma gondii, EBV, CMV, JCV, Cryptococcus neoformans, *Aspergillus* spp., or *Mycobacerium tuberculosis*) [6].

The prevalence of past toxoplasmosis infection (confirmed with serologic testing) in HIV-infected patients varies by population, race, and risk factors. The reported prevalence rates range between 3% and 97% [7] with a rate of 46.12% found in a large meta-analysis of 2101 patients [8].

Before the era of antiretroviral drugs, toxoplasmic encephalitis (TE) used to be the most common focal lesion found in these patients. However, it still remains one of the most frequent causes of morbidity and mortality in patients with AIDS [9]. Therefore, anti-toxoplasma IgG and IgM testing remains the diagnostic standard in every newly diagnosed HIV patient [10]. The CNS lesions typical of toxoplasmosis are round, with a ring-like peripheral contrast enhancement and a mass effect. The diagnosis is confirmed with a positive CSF PCR assay for *T. gondii*. Treatment involves a several-month long regimen of oral sulfasalazine with pyrimethamine or alternative regimens with tritmetoprim/co-trimoxazole [11].

Primary central nervous system lymphoma (PCNSL) is a common malignancy seen in patients with AIDS. It is classed as a B-cell lymphoma, its aetiology is linked to Epstein–Barr virus (EBV) infection, and its subacute symptoms can develop within weeks. The lesions resemble those of toxoplasmosis, being focal, round, with peripheral contrast enhancement and a mass effect causing oedema of the adjacent tissue. About 50% of PCNSL are found post mortem during the autopsy [12]. 

Patients with cerebral cryptococcosis typically present with progressively worsening, persistent headaches. The imaging findings include diffuse brain lesions. The diagnosis is ultimately confirmed with positive cerebral fluid culture, with reactive antigen testing seen as an indirect confirmation only. The treatment involves a long-term oral antifungal regimen of amphotericin with flucytosine [13].

Central nervous system tuberculosis is a significant opportunistic bacterial disease among severely immunocompromised patients living with HIV. It may take the form of tuberculous meningitis (TBM) or a focal lesion related to tuberculoma. The most common symptoms are: drowsiness, headache, nausea, vomiting and stiff neck, paresis, cerebellar symptoms, disturbed consciousness, and seizures. Complications may include palsy of the cranial nerves, CNS ischemia with symptoms of a stroke, or hydrocephalus. An additional significant problem is the multi-drug resistance of mycobacteria causing tuberculosis and the risk of developing an immune reconstruction inflammatory syndrome (IRIS) with a severe, potentially fatal course [14,15].

Progressive multifocal leukoencephalopathy (PML) is caused by the JC (John Cunningham) virus. It is an opportunistic infection of oligodendrocytes and astrocytes causing demyelinating changes seen in magnetic resonance imaging of the brain. The structural and functional neurological impairment progresses gradually, being virtually irreversible and leading to death, due to the lack of effective treatment [16].

Overall, prognosis and therapeutic options are significantly affected by the severity of immunodeficiency and the time of diagnosis. It is thought that up to 20% of HIV-related focal brain lesions (HFBL) in patients with HIV/AIDS require invasive procedures such as biopsy to confirm the diagnosis [17].

Despite the availability of highly effective antiretroviral therapies in developed countries, such as Poland, HIV infections are still detected too late. Unfortunately, this often happens at the stage of symptomatic AIDS in late presenters, defined as a new HIV infection with a low CD4 count (<350 cells/μL) or the onset of an AIDS-defining clinical condition, regardless of the CD4 count. Neurological symptoms are very often the only symptoms in patients with AIDS. CNS pathologies are particularly often detected in HIV-infected patients who discontinue antiretroviral treatment for a number of reasons including but not limited to active substance abuse, alcohol addiction, or coexisting psychiatric disorders. Nevertheless, the causes of neurological disorders in these patients are complex and not only HIV-related.

Late presentation and concomitant CNS lesions are common causes of death in HIV-positive patients.

## 2. Materials and Methods

The aim of our analysis was to evaluate the characteristics of the group of patients with HIV infection who died during hospitalization at our ward, including the subgroup of patients presenting neurological symptoms during hospitalization, as well as to analyse the causes of death taking into account AIDS-related or non-AIDS-related diseases. In addition, we wanted to summarize the epidemiology of lesions in the CNS taking into account autopsy results.

Hospital records of 113 patients with HIV infection confirmed with positive Western blot and/or PCR assay, who died during hospitalization at Infectious Diseases Wards and Intensive Care Unit of the Regional Specialist Hospital in Wrocław, Poland, between 2009 and 2018 were included in this retrospective analysis alongside autopsy reports where available (*n* = 31, 27.5%).

The study group was divided into two subgroups based on clinical presentation into patients with CNS manifestation (including headache, impaired consciousness, impaired speech, hemiparesis, seizures) identified during the hospitalization (*n* = 62; 55%; subgroup A) and those without clinically detectable CNS manifestation (*n* = 51, 45%; subgroup B).

## 3. Results

Demographic characteristics and possible routes of HIV transmission in both subgroups are shown in Table 2 and Table 3. Men constituted the majority in the study cohort (*n* = 93, 82%) and in both subgroups (*n* = 52, 84% in subgroup A vs. *n* = 41, 80% in subgroup B). The age range was 20–65 years, with the identical mean age of 40.4 years and median age of 39 years in both subgroups. The largest proportion of patients (44/113 persons, 39%) were within the age range of 30–39 years. The intravenous drug use (IVDU) was the most common (*n* = 88, 78%) possible route of HIV transmission (42% in subgroup A vs. 35% in subgroup B). HIV transmission through heterosexual, bisexual, and homosexual (MSM) contacts was confirmed in 15 (13%) patients (7% in subgroup A vs. 8% in subgroup B). The MSM is currently the predominant HIV transmission route in Poland [18]. 

Past toxoplasmosis infection was confirmed by serological testing in 50 (44.25%) patients, including 27 (43.55%) from subgroup A and 13 (25.5%) from subgroup B. 

The CD4 count data was only available for 102 patients. The mean CD4 count in that group, calculated for the last determination before death, was 131 cells/mm^3^. The median CD4 count was 59 cells/mm^3^. The mean and median CD4 nadir CD4 was 88.9 and 34 cells/mm^3^, respectively. Interestingly, the mean (112 vs. 157 cells/mm^3^) and median (52 vs. 82 cells/mm^3^) CD4 counts were lower in subgroup A than in subgroup B. These differences, however, were not significant (*p* = 0.2119) (Table 4).

There were 41 (36%) late presenters, including 25 patients (40.3%) in subgroup A and 16 patients (31.4%) in subgroup B. A total of 61 patients (54%) were on antiretroviral therapy (ART), including 28 patients (46.8%) in subgroup A and 23 patients (45%) in subgroup B. A total of 19 patients (16.8%) from both subgroups had ART commenced de novo during their studied hospitalization. This low percentage of patients already receiving or started on antiretroviral therapy was due to late detection of infection (very serious general condition on admission), as well as previous ART discontinuation due to non-compliance by uncooperative patients with active substance abuse, alcohol addiction or mental health problems.

The mean time of death from the diagnosis of HIV infection was 5.32 years (range: 1 day–26 years), including the mean of 4 years in subgroup A and the mean of almost 7 years in subgroup B.

The HBsAg was positive in 11 (9.7%) patients including 7(11.3%) patients in subgroup A and 4 (7.8%) patients in subgroup B. Latent HBV infection (only anti-HBc total reactive) was detected in 57 (50.44%) patients, including 31 (50%) patients in subgroup A and 26 (51%) patients in subgroup B. A total of 68 (60%) patients tested positive for anti-HCV Ab, including 38 (61.3%) in subgroup A and (30, 58.5%) in subgroup B. The reactive HBsAg, anti-HBc total, and anti-HCV were not significant predictors of CNS manifestation (*p* = 0.8348). The HCV and/or HBV viral load data was unavailable as most patients did not have it determined during their final hospitalisation. The high rates of active and past infections with hepatotropic viruses can be explained by patients’ high-risk behaviours, in particular their intravenous drug use. Our findings are similar to those of other researchers who reported the prevalence rate of HBV infection as 6–14%, and of HCV infection as 25–30% in people living with HIV (PLWH), as compared to 72–95% in HIV-positive IVDUs [19]. 

The patient’s condition on admission was extremely serious or serious in 59 (52%) cases, fair to serious in 24 (21%) cases, fair in 25 (22%) cases, and good in 5 (4%) cases, which most likely affected the mean duration of hospital stay. The mean length of inpatient treatment in the entire cohort was 20 days (median 15 days). There was no association between the patient’s general condition on admission and the CNS manifestation (*p* = 0.1713). The comorbidity rates were high-110 (97%) patients had three or more comorbidities (Table 4).

The mean length of inpatient treatment was slightly longer in subgroup A than in subgroup B (almost 21 days vs. almost 18 days). Cardiopulmonary resuscitation was only performed in nine (8%) patients after a sudden cardiac arrest. This included two cases in subgroup A and seven cases in subgroup B, and only resulted in restoring the spontaneous circulation in one case (that patient was then transferred to the ICU, where he eventually died). Such a low number of cardiopulmonary resuscitation attempts can be explained by the otherwise critical condition of the patients, which was not conducive to their survival with maintained quality of life.

The autopsy was performed in 31 (27.4%) cases. The cause of death was determined to be HIV-related/AIDS-associated in 88 (78%) patients, as per the ICD-10 preliminary and secondary diagnostic codes. A similar percentage (76.4%) of AIDS-related deaths was reported in a Chinese study [20]. This rate is significantly higher than the 50% rate reported elsewhere for AIDS-related causes of death in deceased HIV-positive patients. For comparison: 47% in a survey conducted in France, 58% in England, and 39% in Japan [21,22,23]. It should be emphasized that over the past decades an increase has been observed in deaths from non-AIDS-defining diseases among patients living with HIV [24]. However, our group of patients included hospitalized ones, hence these results may be slightly overestimated when compared to the analysis of deaths in all HIV-infected patients. The causes of death of the remaining patients included cirrhosis with end-stage liver disease, urosepsis, bacterial pneumonia, and malignancy. 

A further detailed retrospective analysis of subgroup A (*n* = 62) followed. Out of 62 patients with CNS manifestation, a neurological consultation was requested in 15 (24.2%) cases, a CT scan of the head was requested in 45 (72.6%) cases, an MRI of the head was requested in 23 (37.1%) cases, a lumbar puncture with CSF analysis was carried out in 35 (56.45%) cases, EEG was requested in 1 (1.6%) case. None of the patients had stereotactic biopsy of the brain performed. A finding of interest was that almost a half of our patients (*n* = 26; 42%) with evident neurological symptoms were admitted to our hospital as a transfer from different external departments, including neurology (9), internal diseases (6), psychiatry (6), or A&E (5). The remaining cases (*n* = 36, 48%) were direct admissions, most likely due to known HIV infection.

A confirmed diagnosis and initial cause of death were only determined in 19 (30.65%) cases, including:− CNS toxoplasmosis-diagnosis based on CSF PCR positive for T. gondii (1 case); − CNS cryptococcosis-diagnosis based on CSF culture positive for Cryptococcus neoformans (10 cases);− CNS tuberculosis-diagnosis based on CSF PCR positive for Mycobacterium tuberculosis (3 cases);− Bacterial neuroinfection-diagnosis based on CSF PCR positive for other pathogens (*S. aureus* and *S. pneumoniae*, 1 case of each);− PCNSL-brain specimen collected during the autopsy was sent for histology evaluation which confirmed the diagnosis (2 cases);− Stroke-CT of the head showed features in keeping with new haemorrhagic stroke (1 patient).

The probable causes of neurological symptoms occurring in 43 (69.35%) patients included toxoplasmosis, PML, PCNSL, cryptococcosis, and haemorrhagic stroke—all diagnosed based on typical radiological features seen in CT/MRI of the head. Furthermore, HIV-unrelated conditions, such as respiratory failure, end-stage chronic kidney disease, or hepatic encephalopathy secondary to end-stage liver disease, were identified as causes of neurological symptoms (mostly disorientation or impaired consciousness) in 20 patients (Table 5). 

We further excluded patients with known metabolic disorder (*n* = 20), yielding the following causes of CNS presentation in the remaining patients (*n* = 42):

Mass effect due to focal brain lesions (toxoplasmosis, lymphoma, tuberculoma) in 15 (35.7%) cases;

Meningitis-cryptococcosis, neuroinfections, tuberculous meningitis (TBM) in 18 (42.86%) cases;

White matter disease (PML) in 7 (16.67%) cases;

Cerebrovascular event (stroke) in 2 (4.76%) cases.

Autopsy was performed in 18 deceased patients from subgroup A. Then we assessed the compliance between the death certificate cause of death and autopsy cause of death (Table 6). 

An interesting and important finding from this part of the study is that only histology evaluation of brain lesions enabled a definitive diagnosis. This can be illustrated with examples of patients treated for suspected CNS toxoplasmosis who were diagnosed with CNS lymphoma post mortem or those with suspected brain tumours finally diagnosed with toxoplasmosis post mortem. This finding highlights the need for a wider use of stereotactic biopsy of the brain in patients with focal brain lesions [17], as prompt diagnosis followed by appropriate treatment may increase the patient’s chances of survival. 

There were most likely cases of IRIS; however, the retrospective data did not allow for reliable evaluation; therefore, we did not mention it in our work.

## 4. Discussion

The current analysis of the causes of death was carried out in a relatively small cohort out of all HIV-positive patients treated in our centre. The currently available antiretroviral therapies, alongside a regular thorough outpatient monitoring ensure effective control of the disease, significantly reducing the risk of progression to AIDS and the development of AIDS-defining CNS disease. 

The most common AIDS-defining CNS diseases in our cohort were toxoplasmosis and cryptococcosis. A retrospective analysis published by Brazilian researchers yielded similar findings of the most frequent aetiology of CNS lesions based on autopsy reports of 284 HIV-positive patients, identifying them as toxoplasmosis, cryptococcosis, bacterial infection, and HIV encephalitis (aka HIV-associated neurological disorder, HAND) [19].

Interestingly, metabolic disorder unrelated to any specific CNS pathology (e.g., end-stage liver disease with hepatic encephalopathy or impaired consciousness in patients with respiratory failure observed) was a cause of neurological manifestation in 20 (32.25%) patients with AIDS in our study. 

Even the autopsy and histology tissue evaluation, which should theoretically provide unambiguous answers to all diagnostic doubts, are not always successful at determining a definitive diagnosis.

## 5. Conclusions

Treatment success in patients with AIDS largely depends on early detection of HIV infection.A comprehensive neurological assessment should always be performed in each HIV-positive patient/patient with AIDS, even in the absence of evident CNS manifestation, as timely correct diagnosis improves the chances of therapeutic success.CNS manifestation is a negative prognostic factor in HIV-positive patients/patients with AIDS.Even in 21st century, with the availability of high-end diagnostic methods (laboratory tests, molecular techniques, diagnostic imaging, histology), a number of deaths of HIV-positive patients/patients with AIDS cannot be ultimately explained with a certain clinical diagnosis.There seems to be a justified need for routine HIV testing in patients admitted to neurology departments.

## Figures and Tables

**Table 1 medicina-58-00270-t001:** Central nervous system disorders in HIV patients (CNS-D)–modified by the authors.

1. Primary infection of the brain by HIV:	a. AIDS dementia complex = HIV associated dementia complex = HIV encephalitis = AIDS encephalopathy	
2. Opportunistic infectious:	A. Parasites:	a. Toxoplasma gondii
		b. Cysticercosis (Taenia solium)
	B. Fungal:	a. Cryptococcus neoformans
		b. Candida albicans
		c. Aspergillosis
		d. Coccidioidomycosis
		e. Mucormycosis
		f. Histioplasmosis
	C. Bacterial:	a. Mycobacterium tuberculosis
		b. Mycobacterium avium-intracellulare
		c. Treponema pallidum
		d. Other causes of purulent meningitidis (Streptococcus pneumoniae, Staphylococcus aureus, Neisseria meningitidis, Listeria monocytogenes et.al)
	D. Viral:	a. JCV (causing PML-Progressive multifocal leucoencephalopathy)
		b. other: CMV, VZV, HSV-1, HSV-2
3. Neoplasm:	a. Primary central nervous system lymphoma (PCNSL)	
	b. Kaposi sarcoma	
	c. Primary tumors of the brain	
	d. Brain metastases	
4. Complications of systemic disorders:	a. Metabolic encephalopathy (due to hypoxia, drug, electrolyte imbalance, liver failure, kidney failure, respiratory failure	
	b. Cerebral ischemic or hemorrhagic infarction (stroke)	

**Table 2 medicina-58-00270-t002:** Demographic characteristics of HIV-positive/AIDS patients (*n* = 113)—by age and sex.

		With CNS Manifestation (Subgroup A)	Without CNS Manifestation (Subgroup B)	Total
Sex	Women	10 (16%)	10 (20%)	20 (18%)
	Men	52 (84%)	41 (80%)	93 (82%)
Age (years)	20–29	7 (11.3%)	7 (13.7%)	14 (12%)
	30–39	25 (40.3%)	19 (37.25%)	44 (39%)
	40–49	20 (32.3%)	17 (32.7%)	37 (33%)
	50–60	9 (14.5%)	6 (11.5%)	15 (13%)
	>60	1 (1.6%)	2 (3.85%)	3 (3%)
Mean age (years)		40.4	40.4	40.4
Median age (years)		39	39	39
Total		62 (55%)	51 (45%)	113 (100%)

**Table 3 medicina-58-00270-t003:** Demographic characteristics of HIV/AIDS patients (*n* = 113)—by route of transmission.

Route of Transmission	Sex	With CNS Manifestation	Without CNS Manifestation	Total
IVDU	Women	7	8	15
	Men	41	32	73
	Total:	48 (42%)	40 (35%)	88 (78%)
MSM	Women	0	0	0
	Men	3	4	7
	Total:	3 (3%)	4 (4%)	7 (6%)
HTX	Women	2	2	4
	Men	1	1	2
	Total:	3 (3%)	3 (3%)	6 (5%)
BI	Women	0	0	0
	Men	1	1	2
	Total:	1 (1%)	1 (1%)	2 (2%)
Vertical	Women	1	0	0
	Men	0	0	0
	Total:	1 (1%)	0 (0%)	1 (1%)
Unknow	Women	0	0	0
	Men	6	3	9
	Total:	6 (5%)	3 (3%)	9 (8%)
Total		62	51	113 (100%)

**Table 4 medicina-58-00270-t004:** Subgroup characteristics-selected clinical and laboratory parameters.

		With CNS Manifestation-Subgroup A (*n* = 62)	Without CNS Manifestation-Subgroup B (*n* = 51)	Total: (*n* = 113)
Most recent CD4 count	mean	112	157	131
	median	52	82	59
CD4 nadir	mean	89	88	89
	median	24	50	24
Late presenters		25 (40.3%)	16 (31.4%)	41 (36%)
ART commenced during hospitalisation		14 (22.6%)	5 (9.8%)	19 (16.8%)
ART total		28 (46.8%)	23 (45%)	61 (54%)
Mean time of death from HIV infection diagnosis		4.05 years	6.87 years	5.32 years
HBsAg		7 (11.3%)	4 (7.8%)	11 (9.7%)
Latent HBV infection (only anti-HBc total reactive)		31 (50%)	26 (51%)	57 (50.44%)
Anti-HCV reactive		38 (61.3%)	30 (58.8%)	68 (60%)
Patient general condition	Extremely serious and serious	33 (53.2%)	26 (51%)	59 (52%)
	Fair to serious	16 (25.8%)	8 (15.7%)	24 (12%)
	Fair	12 (19.4%)	13 (25.5%)	25 (22%)
	Fairly good	1 (1.6%)	4 (7.8%)	5 (4%)
Mean hospital stay (days)		20.95	17.8	20
CPR		2 (3.2%)	7 (13.7%)	9 (8%)
Autopsy		18 (29%)	13 (25.5%)	31 (27.4%)
HIV-related cause of death		51 (82%)	37 (72.5%)	88 (79%)

**Table 5 medicina-58-00270-t005:** CNS pathologies identified in hospital records of HIV-positive patients presenting with neurological symptoms who died in 2009–2019 LD-likely diagnosis; CD-confirmed diagnosis (*n* = 62).

	With Neurological Symptoms	Without Neurological Symptoms	Total
Late presenters	25	16	41 (36%)
ARV initiated at the hospital	14	5	
ARV—total	29	23	71 (63%)
Mean length from HIV diagnosis to death	4.05 years (1 day; 20 years)	6.87 years (2 days; 26 years)	5.32 years
HBsAg	7	4	11 (38.1%)
Subclinical HBV infection	31	26	57 (50.44%)
Anti HCV	38	30	68 (60%)
Mean hospital stay	20.95 days	17.8 days	20 days
Resuscitation	2	7	9 (8%)
Autopsy	18	13	31 (27.4%)
HIV-related cause of death	51	37	88 (79%)

**Table 6 medicina-58-00270-t006:** Diagnosis of neurological lesions in HIV-infected patients who died between 2009 and 2019, in our own material. LD—likely diagnosis; CD—confirmed diagnosis (*n* = 62).

			LD	CD	Total
1. Primary infection of the brain by HIV	a. HIV encephalitis		2	0	2 (3.2%)
2. Opportunistic infectious	A. Parasites:	a. Toxoplasma gondii	10	1	11 (17.7%)
		b. Cysticercosis	0	0	0
	B. Fungal:	a. Cryptococcus neoformans	0	10	10 (16.1%)
		b. Candida albicans	0	0	0
		c. Aspergillosis	0	0	0
		d.Coccidioidomycosis	0	0	0
		e. Mucormycosis	0	0	0
		f. Histioplasmosis	1	0	1 (1.6%)
	C. Bacterial:	a. Mycobacterium tuberculosis	0	3	3 (4.8%)
		b. Mycobacterium avium-intracellulare	0	0	0
		c. Treponema pallidum	0	0	0
		d. Other causes of purulent meningitidis	0	2	2 (3.2%)
	D. Viral:	a. JCV	7	0	7 (11.3%)
		b. other: CMV, VZV, HSV-1, HSV-2	0	0	0
3. Neoplasm	a. PCNSL		2	2	4 (6.45%)
	b. Kaposi sarcoma		0	0	0
	c. Primary tumors of the brain		0	0	0
	d. Brain metastases		0	0	0
4.Complications of systemic disorders	a. Metabolic encephalopathy		20	0	20 (32.25%)
	b. Stroke		1	1	2 (3.2%)
	Total		43 69.35%	19 30.65%	62

## Data Availability

Data supporting reported results can be provided upon request from the corresponding author.

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
