# Peer review of "Analysis of Deaths among HIV-Infected Patients Hospitalized in 2009–2018 in Main Centre of Infectious Disease in Region of Lower Silesia in Poland, Detailing Lesions in the Central Nervous System"

_medicina, 2022, doi:10.3390/medicina58020270_

Round 1
Reviewer 1 Report
Line 111-112 – (I and II) - information is not required
Line 177-182 – not relevant for the study
Table 6 – is not relevant, possibly only diagnosed cases could remain
Please note if there have been cases of immune reconstitution in patients with neurological manifestations who have recently received antiviral therapy.
there is a connection between the route of transmission (IVDU) and the CNS manifestations?
272-273 - where does this conclusion come from?
which are the negative predictors for the occurrence of CNF manifestations in patients with HIV and AIDS
which are the negative predictors for the evolution towards death in patients with CNS manifestations
please insert de reference
Streinu-Cercel A, Săndulescu O, Poiană C, Dorobanțu M, Mircescu G, Lăzureanu VE, Dumitru IM, Chirilă O, Streinu-Cercel A, Extended Consensus Group. Consensus statement on the assessment of comorbidities in people living with HIV in Romania. GERMS 2019;9(4):198-210. doi: 10.18683/germs.2019.1178.

Author Response
Dear Reviewer
On behalf of myself and on behalf of the other co-authors, I would like to thank You for the positive comments and remarks to our modest work.
We are greatly encouraged by this opinion.
In response to Your comments:
- Line 111-112 – (I and II) - information is not required
We strongly agree with this remark, information was deleted.
- Line 177-182 – not relevant for the study
Thank you for your attention; however, we believe that this is quite interesting information regarding the failure to undertake resuscitation procedures among the PLWH.
If, of course, You find this fragment irrelevant again, we will, of course, adapt to Your opinion.
- Table 6 – is not relevant, possibly only diagnosed cases could remain
We kindly ask you to include and leave this table at work due to its interesting results and a significant role in the results of the entire work in our opinion
- Please note if there have been cases of immune reconstitution in patients with neurological manifestations who have recently received antiviral therapy.
There were most likely cases of IRIS, however, the retrospective data did not allow for reliable evaluation; therefore, we did not mention it in our work.
Nevertheless, this is a very valid remark for which we thank and we will certainly pay attention to it in the next manuscripts.
- there is a connection between the route of transmission (IVDU) and the CNS manifestations?
Probably not, we did not undertake such an evaluation
- 272-273 - where does this conclusion come from?
Based on the life expectancy of patients in the group with neurological symptoms vs. no neurological symptoms.
- which are the negative predictors for the occurrence of CNF manifestations in patients with HIV and AIDS and which are the negative predictors for the evolution towards death in patients with CNS manifestations
First of all, an undiagnosed HIV infection. Leading to profound immunodeficiency and a low CD4 T cell count, which is reflected in our results; moreover, according to the literature, adherence disorders and discontinuation of ART therapy are also still present; resistance to ART drugs and IRIS.
- please insert de reference
Streinu-Cercel A, Săndulescu O, Poiană C, Dorobanțu M, Mircescu G, Lăzureanu VE, Dumitru IM, Chirilă O, Streinu-Cercel A, Extended Consensus Group. Consensus statement on the assessment of comorbidities in people living with HIV in Romania. GERMS 2019;9(4):198-210. doi: 10.18683/germs.2019.1178.
It was added (position 24 in the bibliography)
We also focus on making small corrections to the comments from the second reviewer.
We believe that we will be able to publish our work after the improvements made.
Yours sincerely
Justyna Janocha-Litwin
Reviewer 2 Report
HIV-AIDS disease is a public health problem worldwide. Damage to the central nervous system in the HIV-infected patient raises significant issues of sustained etiological diagnosis and treatment management. Moreover, it is a substantial cause of mortality, as the authors have shown. Therefore, research is essential for specialists in different fields such as infectious disease doctors, neurologists, pulmonologists, neurosurgeons, general practitioners. The article`s structure is appropriate; the sections are logical and correspond to the topic. The title and the abstract convey the essential aspects of the research. After reviewing knowledge in the field described in the introduction, the research method and analysis are well explained. The results and the conclusions support the research. The references chosen by the authors are appropriate.
The article can be a good tool in clinical practice.
I would propose to describe in a few words the manifestations of the Central nervous system tuberculosis, which represents one of the most devastating manifestations of TB-HIV patients.
Author Response
Dear Reviewer
On behalf of myself and on behalf of the other co-authors, I would like to thank You for the wonderful, full of positive comments and remarks to our modest work.
We are greatly encouraged by this opinion.
Minor remarks on the significant problem of nervous system tuberculosis in HIV patients have been added.
We also focus on making corrections to the comments from the second reviewer.
Yours sincerely
Justyna Janocha-Litwin